# A Limited Effect of Sub-Tropical Typhoons on Phytoplankton Dynamics

Fei Chai[1,2*], Yuntao Wang[1*], Xiaogang Xing[1], Yunwei Yan[1], Huijie Xue[2,3], Mark Wells[2], Emmanuel Boss[2]

[1] State Key Laboratory of Satellite Ocean Environment Dynamics, Second Institute of Oceanography, Ministry of Natural Resources, Hangzhou, 310012, China

[2] School of Marine Sciences, University of Maine, Orono, ME, 04469, USA

[3] State Key Laboratory of Tropical Oceanography, South China Sea Institute of Oceanology, Chinese Academy of Sciences, Guangzhou, 510301, China

Correspondence: Fei Chai (fchai@sio.org.edu) and Yuntao Wang (yuntao.wang@sio.org.cn)

**Abstract.** Typhoons are assumed to stimulate primary ocean production through the upward mixing of nutrients into the ocean surface. This assumption is based largely on observations of increased surface chlorophyll concentrations following the passage of typhoons. This surface chlorophyll enhancement, occasionally detected by satellites, is often undetected due to intense cloud coverage. Daily data from a BGC-Argo profiling float revealed the upper-ocean response to Typhoon Trami in the Northwest Pacific Ocean. Temperature and chlorophyll changed rapidly, with a significant drop in sea surface temperature and a surge in surface chlorophyll associated with strong vertical mixing, which was only partially captured by satellite observations. However, no net increase in vertically integrated chlorophyll was observed during Typhoon Trami or in its wake. In contrast to the prevailing dogma, the result shows that typhoons likely have a limited effect on net primary ocean production. Observed surface chlorophyll enhancements during and immediately following typhoons in tropical and subtropical waters are more likely to be associated with surface entrainment of deep chlorophyll maxima. Moreover, the findings demonstrate that remote sensing data alone can overestimate the impact of storms on primary production in all oceans. Full understanding of the impact of storms on upper ocean productivity can only be achieved with ocean observing robots dedicated to high-resolution temporal sampling in the path of storms.

## 1 Introduction

The western North Pacific Ocean is a highly energetic region on the globe (Gray, 1968) and is where nearly one-third of tropical cyclones originate (Needham et al., 2015). The strong tropical cyclones in this region, referred to as typhoons, are highly dangerous and have caused great loss of life and property throughout history (Frank and Husain, 1971; Dunnavan and Diercks, 1980; Kang et al., 2009; Needham et al., 2015). Typhoons extract their energy from warm surface ocean waters, thus the heat content in the upper ocean (quantified by the sea surface temperature (SST) as the indicator) has a key role in development of typhoons (Emanuel, 1999). Increasing SST in the North Pacific over the past few decades (He and Soden,

2015) coincides with an increase in the number of intense typhoons in the region (Emanuel, 2005; Webster et al., 2005; Vecchi et al., 2007; Kossin, 2018), which has been found to relate to climate change (Mei et al., 2015). The trend draws public attention for its potential influence on increasing climate extremes in this region.

Numerous studies have analyzed the impact of typhoons on upper-ocean conditions (e.g., Sun et al., 2010; Zhang et al., 2018). Higher surface ocean temperatures enhance stratification and thus decrease the nutrient flux, which reduces the ability of typhoons to cool the upper ocean and to elevate the growth of phytoplankton (Zhao et al., 2017). Ocean productivity and carbon sequestration are subsequently reduced, helping to sustain the continued global temperature increase (Balaguru et al., 2016). The high winds and strong energy exchange associated with typhoons (Price, 1981), on the other hand, are suggested
to partially reverse this trend by mixing subsurface cold and nutrient-rich water into the sunlit surface layer (Babin et al., 2004). This results in decreasing SST and enhanced phytoplankton growth at the ocean surface (Platt, 1986, Ye et al., 2013).

The feedback from ocean to typhoon, on the other hand, is important for the development and maintenance of typhoons, as typhoons require extracting energy from the ocean surface (Zheng et al., 2008). The translation speed, e.g., the moving speed
of a typhoon, plays an important role in determining the interaction between the ocean surface and typhoon (Pothapakula et al., 2017). The typhoon can lose energy and become weak when passing over a cold surface, such as regions cooled by the typhoon itself; thus, slow moving typhoons can hardly develop into stronger storms (Lin et al., 2009). On the other hand, the longer the time a typhoon lingers around a certain location the stronger is its local impact (Zhao et al., 2013); the resulting cooling effect further dampens the intensity of the typhoon. Thus, strong typhoons in mid-latitude regions are generally
characterized as fast-moving typhoons (Lin, 2012).

As typhoons propagate, they can drive substantial vertical mixing in the upper ocean (Han et al., 2012). Typhoon induced mixing lasts for approximately one week with an impact from the surface to as deep as 100 m (Price, 1994). Mixing acts to deepen the mixed layer depth (MLD), resulting in a redistribution of ocean surface parameters (Lin et al., 2017). For typhoons
passing over regions with shallow water depth and strong stratification, large ocean surface responses are generally observed (Zhao et al., 2017). At the same time, the typhoon-driven upwelling via the wind stress curl is found to influence depths of 200 m or more (Zhang et al., 2018). The relative impact of mixing and upwelling has been compared in many former studies, though their conclusions largely vary. For example, mixing is reported to be much more effective for inducing ocean surface changes compared to upwelling (Jacob, 2000). Lin et al. (2017) compiled the impacts of tropical cyclones for the upper 1000
m in the Northwest Pacific and found that the contribution of mixing is dominant only in the surface layer that is shallower than 35 m.

There is a strong linkage among the stratification in the upper ocean, chlorophyll distribution, and nutricline in mid-latitude regions, where any intensification of vertical mixing often leads to a surge in nutrient flux to surface waters. However, the

same linkage is not necessarily true in tropical and subtropical regions, where a two-stratum sunlit "surface" layer forms. Here, the wind-mixed layer comprises only the upper (shallow) portion of the sunlit and nutrient-depleted surface layer, with the former being much shallower than the latter (Du et al., 2017). In this case, the photic zone extends far below the typical wind-mixed layer, and a deep chlorophyll maximum (DCM) forms at the top of nutricline and may exist at twice or more the depth of the wind-mixed layer (Letelier et al., 2004; Cullen 2015; Gong et al., 2017; Pan et al., 2017). The question then is whether

the energy transfer from typhoons generates sufficient mixing to "break" through both the base of the wind-mixed layer as well as the deeper nutricline, thereby transferring new nutrients into the photic zone. Typhoons induced upwelling can transport nutrient-enriched waters into the photic zone, and has been shown to enhance subsurface chlorophyll concentrations in the South China Sea (Ye et al., 2003). However, the question remains whether similar processes will occur for the subtropical oceans where nutricline is much deeper.


In addition to the intensive wind field, typhoons are also associated with intensified rainfall and cloud cover (Liu et al., 2013), which can substantially contaminate satellite observations. Satellite-based studies occasionally capture the ocean surface features during the passage of a typhoon and offer more data in the wake following typhoons (Chang et al., 2008). Remote sensing data revealed that the SST rapidly decreases during typhoon passage, whereas chlorophyll can increase afterwards

(Chen and Tang, 2012; Zhao et al., 2017). It was suggested that the delayed response of surface chlorophyll is related to the time needed for phytoplankton to exploit the increased nutrient concentrations and accumulate (Chen et al., 2014). The time for phytoplankton growth depends on the phytoplankton species, e.g., at least three days and five days are respectively required for diatoms and small phytoplankton to accumulate significantly after nutrient infusion (Pan et al., 2017).

Typhoon induced ocean responses largely vary in intensity and even in magnitude of changes depending on the typhoon's features and pre-typhoon ocean state. Slow, strong typhoons appear to be favorable for inducing an oceanic response (Lin et al., 2017), which can be amplified by pre-existing cyclonic eddies (Sun et al., 2010). Bauer and Waniek (2013) showed that individual typhoons may more than double the surface phytoplankton biomass, though subsurface effects could not be quantified. On the other hand, Zhao et al. (2008; 2017) found that only strong and slow-moving typhoons impart sufficient

energy to increase surface chlorophyll concentrations. The diverse outcomes observed among typhoons and upper ocean interactions (Chang et al., 2008; Chen and Tang, 2012; Balaguru et al., 2016), weighted heavily by surface-only satellite observations, illustrate the incomplete understanding of how typhoons may affect the primary productivity in the present and future ocean. In this study, the vertical sections obtained with a biogeochemical-Argo (BGC-Argo) profiling float are used to describe the entire upper ocean responses to the passage of a strong typhoon in oceanic subtropical waters, offering novel

insights that detangle the underlying physical and biological dynamics.

## 2 Methods

### 2.1 Data used in this study

The BGC-Argo profiling float (ID: 2902750), an unmanned observational platform (Bishop et al., 2002; Boss et al., 2008; Claustre et al., 2010; Mignot et al., 2014; Chacko, 2017), was deployed by the State Key Laboratory of Satellite Ocean Environment Dynamics (SOED) of China in early September of 2018 in the Northwest Pacific (South of Japan). It was equipped with a CTD (SBE41CP manufactured by Seabird) measuring temperature and salinity and an optical sensor package (ECO Triplet manufactured by WET Labs) measuring chlorophyll-a concentration (Chla; $mg/m^{-3}$), fluorescent dissolved organic matter (FDOM; ppb), and particulate backscattering coefficient at 700 nm (bbp-700; $m^{-1}$). Measurements were made every night (around 22:00 local time) to avoid in vivo fluorescence nonphotochemical quenching. Vertical resolution of measurements was ~ 1 m in the upper 1000 m.

Float data was quality controlled following the requirement of the BGC-Argo Program (Schmechtig et al., 2016) before uploading to the Argo global data assembly center (GDAC). Data used in this study are available from the Coriolis GDAC FTP server (Argo, 2020). Chla and bbp (700) sections were smoothed with a 5-point median filter and their factory-calibrated dark counts for backscattering and Fchl (48 and 48, respectively) were replaced by the on-float-measured counts (50 and 53), which were measured by the sensors on the float before deployment.

The remote sensing observations of SST and sea surface chlorophyll (SSC) were obtained as the MODIS L3 data onboard NASA's EOS-Aqua satellite. Daily observations with global coverage were available from September 2002 until now. The spatial resolution was 1/24°, and there were no data presenting over land or cloud. Satellite observed information near BGC-Argo was calculated by spatially averaging over a surrounding circle with a radius of 300 km following Wang (2020), excluding areas within 10 km of land.

Climatological information for the regions was obtained from the World Ocean Atlas (WOA) at a spatial resolution of 1° (Locarnini et al., 2018). The data had an original 57 layers from the surface to 1500 m depth, and it was interpolated to 1 m vertical resolution beforehand. A monthly dataset of the temperature profile was subsequently used to delineate the climatological MLD, and the calculation will be described in the following section.

The typhoon information is obtained from the Japan Meteorological Agency (JMA, http://www.jma.go.jp/jma/indexe.html). The data include the maximum sustained wind (MSW), category, and location (longitude and latitude) of the typhoon center every 6 hours. The category is defined based on the MSW, and the maximum MSW for the studied typhoon is 105 knots (approximately 54 m/s), which is classified as 'class 5- violent typhoon' for the intensity class (agora.ex.nii.ac.jp/digital-typhoon/help/unit.html.en) and category 3 with the Saffir-Simpson scale (www.nhc.noaa.gov/aboutsshws.php). The translation

speed of the typhoon was calculated as the ratio between the spatial distance traveled between two successive typhoon centers in the 6-hour time span.

## 2.2 MLD and mixing-induced changes in temperature and chlorophyll

Following Kara et al. (2000), the estimate of MLD was obtained using a density-based criterion with the increase in density
equivalent to the decrease in temperature by 0.8°C from the ocean surface. The mixed layer temperature (MLT) and mixed layer chlorophyll (MLC) quantify the typhoon induced changes at daily intervals. The calculation correctly reflected the ocean turbulence mixing, where the MLT and MLC were simply the vertically averaged temperature and chlorophyll concentration from the surface to the MLD. We further defined and calculated the mixing-induced change in MLT in a daily interval using the following Eq. (1):

$$\Delta\text{MLT}_\text{m} = \left.\int_0^h \rho_r C_p T_0 dz \middle/ \rho_r C_p h\right. - \left.\int_0^{h_0} \rho_r C_p T_0 dz \middle/ \rho_r C_p h_0\right. = \left.\int_0^h T_0 dz \middle/ h\right. - \left.\int_0^{h_0} T_0 dz \middle/ h_0\right. \tag{1}$$

where $\Delta\text{MLT}_\text{m}$ is the mixing-induced change in MLT, $\rho_r$ is the relative water density, $C_p$ is the specific heat capacity at constant pressure, $h$ is the MLD, $h_0$ and $T_0$ are, respectively, the MLD and temperature from the previous day. The mixing-induced change in MLC was calculated similarly.

## 3 Results

The typhoon Trami was spawned in the tropical Western Pacific in late September 2018 and moved from the tropics to the mid-latitudes around Japan with an overall translation speed of 6.1 ± 6.5 m/s (Figure 1). The daily averaged translation speed was 19.3 ± 6.1 m/s on September 30, indicating that the typhoon was moving faster compared with other times in its lifespan. During this day, typhoon Trami passed over the BGC-Argo float with an average wind speed of 44 m/s and the shortest distance between the typhoon and the float was less than 100 km. As the float drifted with the ocean circulation, its position changed
slightly during the study period. In particular, the float was located at 133.1°E, 30°N on Sep. 25, 133°E, 30.6°N on Sep. 30, and 133.6°E, 31.4°N on Oct. 5. The zonal movement is not prominent, but the meridional shift is approximately 150 km northward. The climatological MLDs for BGC-Argo locations are 28 m, 41 m, and 60 m at approximately 30°N during August, September and October, respectively, and these values change to 30 m, 46 m, and 66 m at approximately 31°N, indicating a slight deepening towards the north.


The float depth vs. time property sections, consistent with remote sensing data of surface properties, show a rapid drop in surface water temperatures above the base of the MLD and warming below it, which is associated with a swift deepening of the mixed layer upon passage of the typhoon (Figure 2a, c). The largest reduction in SST (> 1°C) happened simultaneously

with the arrival of the typhoon, consistent with the local shear-driven vertical mixing mechanism (Sriver and Huber, 2007; Chang et al., 2008; Zhao et al., 2008; Sanford et al., 2011).

The changes in the vertical distribution of chlorophyll were equally striking, with the chlorophyll concentration in the DCM decreasing from 0.8 to 0.4 mg/m$^3$, while near-surface values more than doubled (Figure 2b, d). We observed two surface chlorophyll peak values on September 30 and October 3, at 0.18 and 0.15 mg/m$^3$, respectively. These increases represent changes of 0.13 and 0.1 mg/m$^3$, respectively, above the concentration measured on September 29 before the typhoon approached to the area. The highest value occurred on September 30 when the surface wind peaked around the BGC-Argo float, which supports the vertical mixing mechanism. This enhancement was, however, not captured by satellite remote sensing due to intense cloud cover from the typhoon.

Vertical mixing appeared to be the dominant physical process during the typhoon Trami passage. The MLD deepened dramatically from 40 m before September 29 to 94 m as the wind speed peaked on September 30 (Figure 3a). While the MLT decreased quickly, the integrated ocean heat content (OHC) of the top 150 m remained relatively constant approximately 1.47 J/m$^2$ (Figure 3b), suggesting that upwelling was minimal. The elevated MLC coincided with decreasing concentrations at DCM (Figure 3c). As a consequence, the strong vertical mixing generated no increase in the depth-integrated chlorophyll within the upper 150-m either during or after the passage of Typhoon Trami (Figure 3d). There was no increase measured in particle backscatter coefficient (bbp) that correlates well with phytoplankton carbon (Graff et al., 2015), suggesting there was no significant change in phytoplankton abundance or any shift towards larger phytoplankton that might enhance carbon export.

The temperature above 50 m decreased with the largest reduction near the surface, while the temperature beneath increased simultaneously (Figure 4a). This process was fully captured as Trami approached until six days after its passing. The largest temperature increase happened between 75 and 100 m from September 30 to October 3 when the surface temperature dropped prominently, which indicated the heat exchange between the upper and lower regions of the photic zone. Consistently, the time series of chlorophyll sections depict the typhoon-induced chlorophyll enhancement near the surface and decrease below (Figure 4b). Because the initial vertical gradient of chlorophyll was very low in the upper 50 m, the increase was mainly uniform throughout the upper 50 m. The largest decrease in chlorophyll was observed at approximately 110-m depth where the initial DCM was located. This pattern is consistent with that from shipboard observations after another typhoon event where a strong vertical movement of particles occurred in the subsurface layer rather than at the surface (Ye et al., 2013). Interestingly, both the temperature and chlorophyll increased in the subsurface, e.g., between 50 m and 100 m, during the week before the arrival of the typhoon, presumably due to the MLD is deepening following its seasonal cycle, associated with increasing winds in front of the typhoon path (Figure 3a).

We calculated the mixing-induced daily change in mixed-layer temperature and chlorophyll ($\Delta MLT_m$ and $\Delta MLC_m$, see the definition in Methods section) and compared them with the daily change in SST and SSC from the BGC-Argo float data ($\Delta SST$ and $\Delta SSC$) (Figure 4c). A good agreement can be found between the mixing induced difference and the actual change. Thus, the observed surface change in temperature was predominantly attributed to strong wind mixing during typhoon passage. The calculated $\Delta MLT_m$ of 1.8°C on September 30 was similar with the value of $\Delta SST$ and consistent with the strong mixing scenario. Similarly, there was close agreement between the observed SSC and calculated $\Delta MLC_m$ that matched the peak on September 30, which indicates that strong vertical mixing contributes to the surface chlorophyll increase.

## 4 Discussion

The passage of a typhoon over the float offers a unique opportunity to fully resolve typhoon-induced oceanic responses throughout the upper water column. In contrast to the interpretation of remote sensing data, super Typhoon Trami did not significantly boost phytoplankton biomass or production in the upper ocean The measured increases in near-surface chlorophyll concentrations resulted simply from the redistribution of phytoplankton, originally residing in the deep chlorophyll maximum, across the mixed layer now reaching throughout the photic zone, as summarized in Figure 5. This finding implies that using satellite observations alone would overestimate typhoon-induced changes in primary production unless consideration is given to chlorophyll redistribution in the water column.

Typhoon-induced mixing dominated the variability of surface chlorophyll during the passage of Trami, which led to the redistribution of chlorophyll within the upper ocean (Figure 3c). In many former remote sensing studies, typhoons were thought to boost primary production by transporting nutrients across nutricline into the photic zone (e.g., Chen and Tang, 2012), and the peak of chlorophyll is typically reached a few days after the minimum SST because time is required for the accumulation of phytoplankton (Shang et al., 2008). However, the high-frequency in situ float profiling here shows almost no changes in integrated temperature and chlorophyll concentration over the top 150 m (Figure 3d). Indeed, the upward displacement of the isotherm is only observed for the upper ocean shallower than 100 m (Figure 2a), whereas upwelling is usually found throughout the water column extending more than 200 m (Lin et al., 2017). The results, differ from that observed nearer to shore in the South China Sea where the pycnocline is much shallower and thus more susceptible to becoming incorporated into the wind-mixed layer (Ye et al., 2013), indicate that mixing is much stronger than upwelling. The outcomes suggest the importance of comparing typhoon-induced mixing depth, thermocline, and nutricline when discussing the upper ocean's responses to typhoons. Similarly, stronger typhoon-induced responses are found in pre-existing cyclonic eddies where nutricline is elevated by upwelling (Zheng et al., 2008; Wu and Li, 2018). A net decrease in heat content and increase in primary production can only be achieved if typhoons penetrate the thermocline and nutricline, respectively (Zhang et al., 2018).

As the mixing process dominated the dynamics in ocean surface, the impact of typhoon Trami happens and fades quickly during and after the typhoon, respectively (Figure 2d). Surface chlorophyll concentration usually relaxed to its initial value within the next few days, even weeks, after the typhoon, which was formerly attributed to the consumption of nutrients (Shibano et al., 2011) and grazing (Zhou et al., 2011; Chung et al., 2012). Conversely, chlorophyll biomass rapidly fades in next five days at surface and increased quickly at depth to reestablish the DCM (Figure 2b), indicating grazing would have played a greater role at surface given the new nutrient inputs would have been minimal. By comparison, surface water temperature rose slowly, associated with the gradual shoaling of the MLD and increasing stratification (Figure 2a). This is attributed to weaker solar radiation in the Trami typhoon region compared to stronger tropical solar radiation, which allows the SST and stratification rebound quickly after passage of a typhoon (Gierach and Subrahmanyam, 2008).

The location of the float is approximately 200 km away from land, so it is unlikely that typhoon-induced rainfall on shore might have increased terrestrial nutrient fluxes sufficiently to influence the growth of phytoplankton (Zheng and Tang, 2007). Though typhoon-induced surface advection can extend southward from the coast of Japan for more than 100 km (Yang et al., 2010), there is not significant trend in satellite-derived chlorophyll concentrations across the 10-300 km range from land (Figure 1). The terrestrial impact induced by intensive rainfall is not prominent for the region around the float, which is different from other regions with estuaries, e.g., Pearl River Estuary, where the rainfall associated with a typhoon can substantially increase nutrient concentration in the nearshore region (Zheng and Tang, 2007).

Typhoon Trami is characterized as a strong and fast-moving typhoon with large wind intensity and translation speed (Wu and Li, 2018), which are favorable for inducing prominent changes in surface ocean processes (Pothapakula et al., 2017). There was almost no lingering during its propagation, and the self-induced cooling did not largely impact the typhoon intensity (Glenn et al., 2016). Indeed, the ocean surface cooling and bloom were particularly intense along the right-hand side of the typhoon track where the float was located (Figure 1a). The right side of typhoons often believed to be subject to more energetic mixing (e.g., Babin et al., 2004), although Huang and Oey (2015) showed only weak asymmetry in mixing across storms.

The typhoon-induced oceanic response was observed in a relatively short period, e.g., the rapid change happened within a few days and the recovery took few weeks (Figure 4), but the associated large-scale environmental changes should also be evaluated, especially the change induced by the drifting of the float. The MLD deepened during the period when the float drifted northward, though only a small distance, and followed the seasonal variation in MLD (not shown); thus, both the subsurface temperature and chlorophyll elevated due to intensified mixing (Figure 4a, b). Additionally, the wind stress increased before the arrival of the typhoon, resulting in ahead-of-eye cooling and mixing that has been shown to take place a few days earlier (Glenn et al., 2016; Wang, 2020).

The decrease in SST is a well-known phenomenon associated with typhoons; in comparison, the typhoon-induced variation in chlorophyll concentration is much more complex and varies by case. Increased chlorophyll concentrations are found in 70% of typhoons in the South China Sea (Wang, 2020) and 18% of typhoons in northwest Pacific Ocean (Lin, 2012). These satellite-derived observations of increased surface chlorophyll may be due to enhanced nutrient flux into surface waters from either vertical mixing or enhanced upwelling reaching beyond the nutricline (Zhang et al., 2018), or as shown here, simply mixing of phytoplankton biomass in the DCM throughout the mixed layer (Figure 2). Distinguishing among these mechanisms cannot be done by satellite observations alone, due both to the high cloud coverage associated with typhoons (Chen and Tang, 2012) and the limitation of sampling only the uppermost ocean surface. The novel float dataset here reveals an added complexity to understanding the progressive effects of typhoon intensity on surface water dynamics and phytoplankton production.

The findings here have a broad application for assessing the impact of typhoons on global primary production and carbon cycling (Menkes et al., 2016). A climate-driven increasing trend of super typhoons (Elsner et al., 2008; Sobel et al., 2016) are likely to cause larger biogeochemical alterations in shelf and nearshore regions where nutricline is comparatively shallow, but they will be much less effective in boosting oceanic productivity in regions where the nutricline is deep and thus more sheltered from enhanced vertical transport. Re-examination of past storm events in this context would better quantify changes in primary production, but our findings imply that future increases in storm frequency alone are unlikely to mitigate the declining trend in global phytoplankton biomass resulting from the enhanced stratification due to warming (Boyce et al., 2010; Lin and Chan, 2015).

The standard float profiling cycles of every 5-10 days to extend their operational lifetimes (Johnson and Claustre, 2016); this sampling frequency is too low to capture the daily or weekly variability induced by synoptic weather and other short-term events. Near-daily sampling frequencies, on the other hand, better inform on rapid processes in response to short-term atmospheric perturbations (Xing et al., 2020). Combined with high-frequency remote sensing data, these observations would enable the development of new and more comprehensive conceptual and quantitative models (Terzić et al., 2019) to improve our understanding of how climate drivers will influence ocean primary production and the associated carbon export in future oceans.

## 5 Conclusions

Observational datasets from daily BGC-Argo sections provide a unique opportunity to study the impact of typhoons on the upper ocean structure and productivity in subtropical waters. The daily profiling frequency capture of the rapid response of the ocean surface to typhoon Trami, and the findings show substantial increases in vertical mixing and surface phytoplankton biomass, similar to past studies. Unlike observations in nearshore regions with shallower nutricline, upwelling contributed new nutrients into photic layer and enhanced net growth of phytoplankton. In current study, the results clearly showed that mixing

overwhelmed the dynamics during the passage of typhoon Trami, while the impact of upwelling was much less pronounced. The observed surface phytoplankton bloom is actually attributed to mixing in upper ocean, e.g., redistribution of the DCM throughout the mixed layer, generating no net increase in phytoplankton biomass. Satellite-based studies reporting delayed phytoplankton blooms in the wake of typhoons over oceanic waters, which infer subsurface nutrient infusion, may be biased by cloud cover obstructing surface observations during the passage of a typhoon. Our findings imply that previous assumptions have overestimated typhoon-enhanced phytoplankton production from mixing or upwelling transport of new nutrients into the photic zone, and that much of this inferred new production instead is driven by vertical redistribution of existing biomass from the DCM. The findings here also provide a lesson on how cutting-edge observational platforms such as BGC-Argo floats can be adapted to resolve short-term, episodic events, delivering information that is not fully resolved by traditional observations. Automatically adapting systems, which change the sampling frequency based on the duration of targeted events (Chai et al., 2020), will be important for improving the understanding of weather and climate impacts on the marine system.

*Acknowledgements*. The BGC-Argo float data used for this study can be downloaded from ftp://ftp.ifremer.fr/ifremer/argo. Satellite observation files are available from the database of National Aeronautics and Space Administration (NASA, podaac-tools.jpl.nasa.gov/drive/files/allData/modis). The typhoon information was downloaded from Japan Meteorological Agency (JMA, www.jma.go.jp/jma/indexe.html). The authors appreciate the Argo and BGC Argo program, JMA and NASA for sharing the data.

*Author contributions*. F. C. designed the observational plan, interpreted the data, and wrote the manuscript; Y. W. organized the figures and results, and worked on the draft; X. X. and Y. Y. conducted the data analysis; H. X., M. W., and E. B. provided insightful comments and improved the manuscript.

*Financial support*. This research was supported by the National Key Research and Development Program of China under contract 2016YFC1401600, the National Natural Science Foundation of China under contracts 41730536 and 41890805.

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

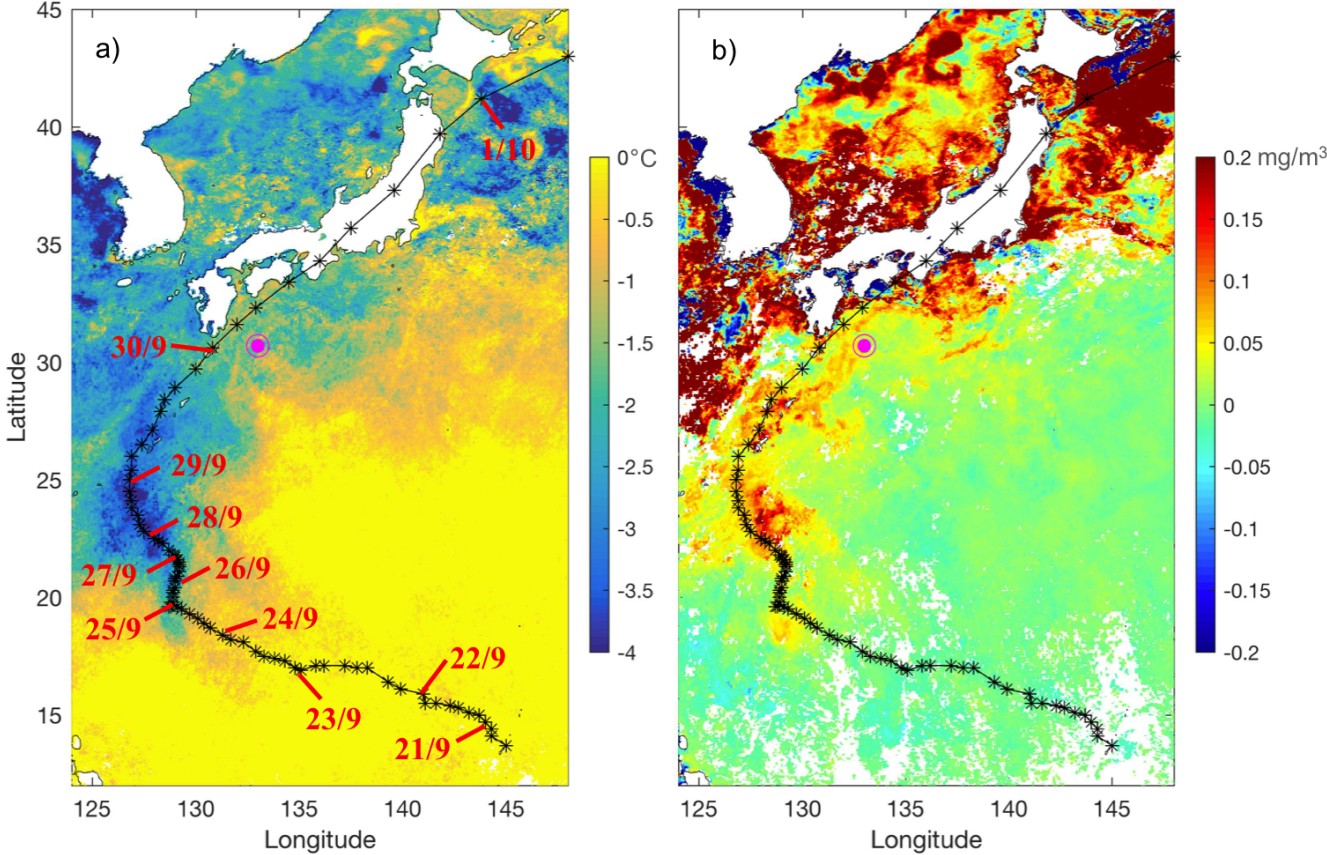

**Figure 1: The differences in averaged sea surface temperature (a) and chlorophyll (b) measured during the period from September 10 to September 30 versus that measured from October 1 to October 20. The remote sensing observations were obtained as the MODIS L3 daily data. The trajectory of Typhoon Trami is shown as a black curve with its central location every 3-hour labeled as**
**asterisks and the date labeled for the location at noon (UTC). The location of biogeochemical-Argo (BGC-Argo) on September 30 is shown as a magenta symbol.**

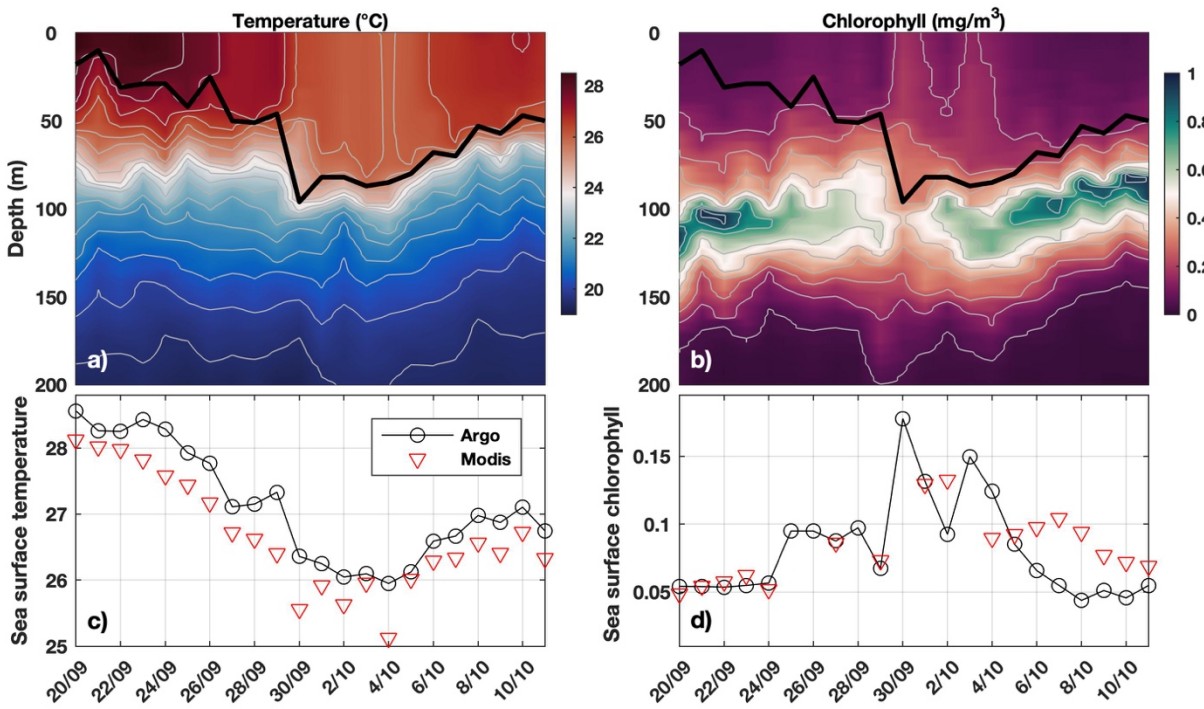

**Figure 2: Sections of temperature (a) and chlorophyll (b) captured by the BGC-Argo; superimposed as the thick black curve is the mixed layer depth (MLD). Comparison between the sea surface temperature (c) and chlorophyll (d) from the BGC-Argo and remote sensing observations. Typhoon Trami passed nearby BGC-Argo on September 30, 2018.**


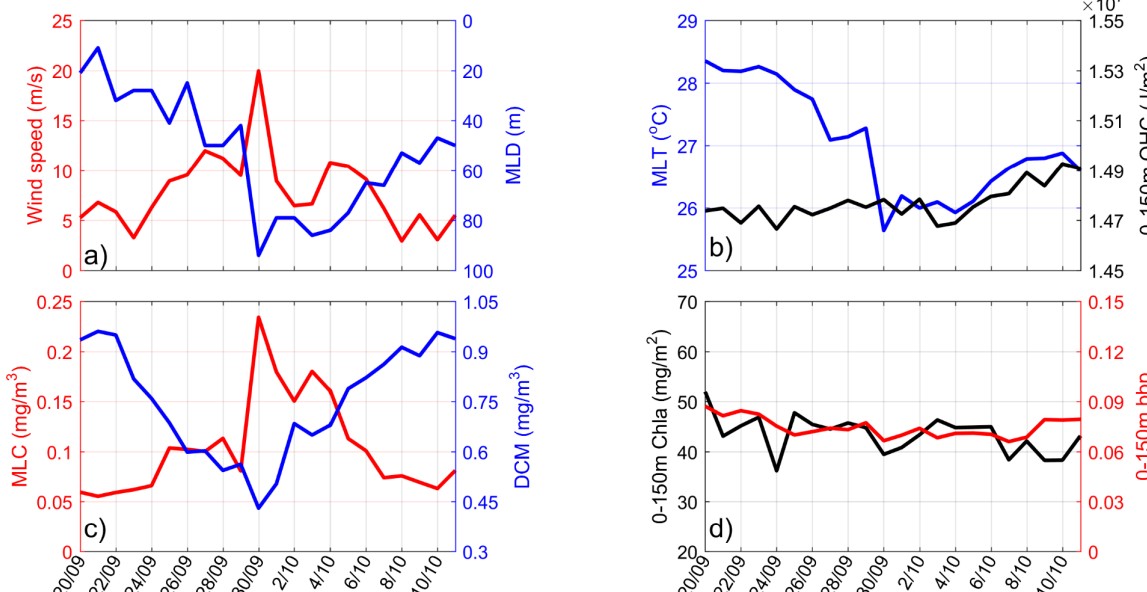

**Figure 3: Time series of (a) the wind speed and MLD at BGC-Argo, (b) mean temperature above the MLD (MLT) and integrated ocean heat content (OHC) between 0 and 150 m, (c) mean chlorophyll above the MLD (MLC) and deep chlorophyll maximum (DCM), and (d) integrated chlorophyll and bbp between 0 and 150 m.**

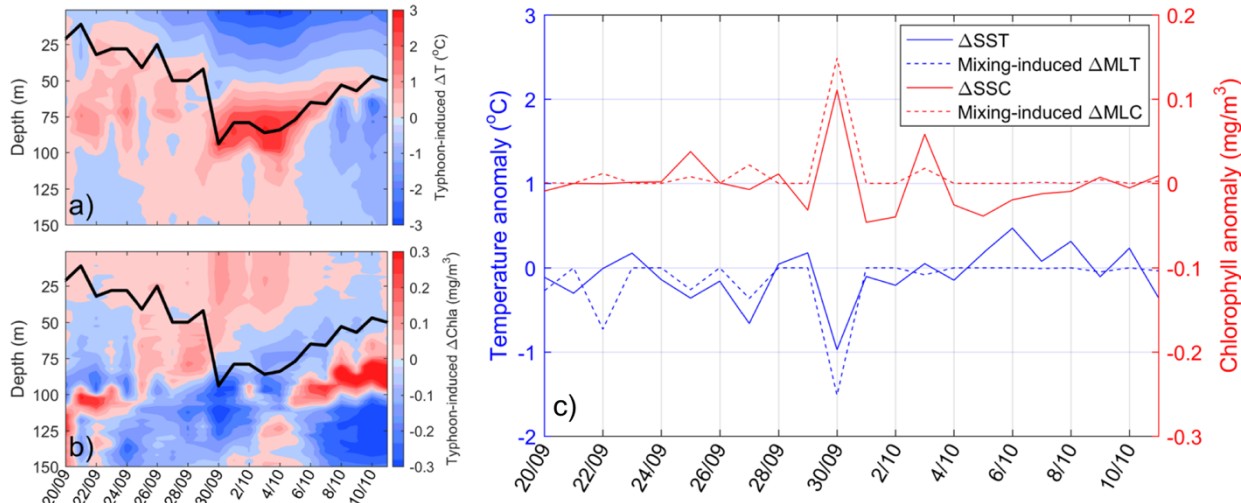


**Figure 4: Sections of typhoon-induced changes in (a) temperature and (b) chlorophyll concentration. The changes were calculated for each day's section by subtracting the averaged section from seven days prior to September 20 (i.e., September 13 to 19). (c) Time series of the change in sea surface temperature (ΔSST, solid blue), mixing-induced change in the MLT (dashed blue), sea surface chlorophyll concentration (ΔSSC, solid red) and mixing-induced change in the MLC (dashed red).**


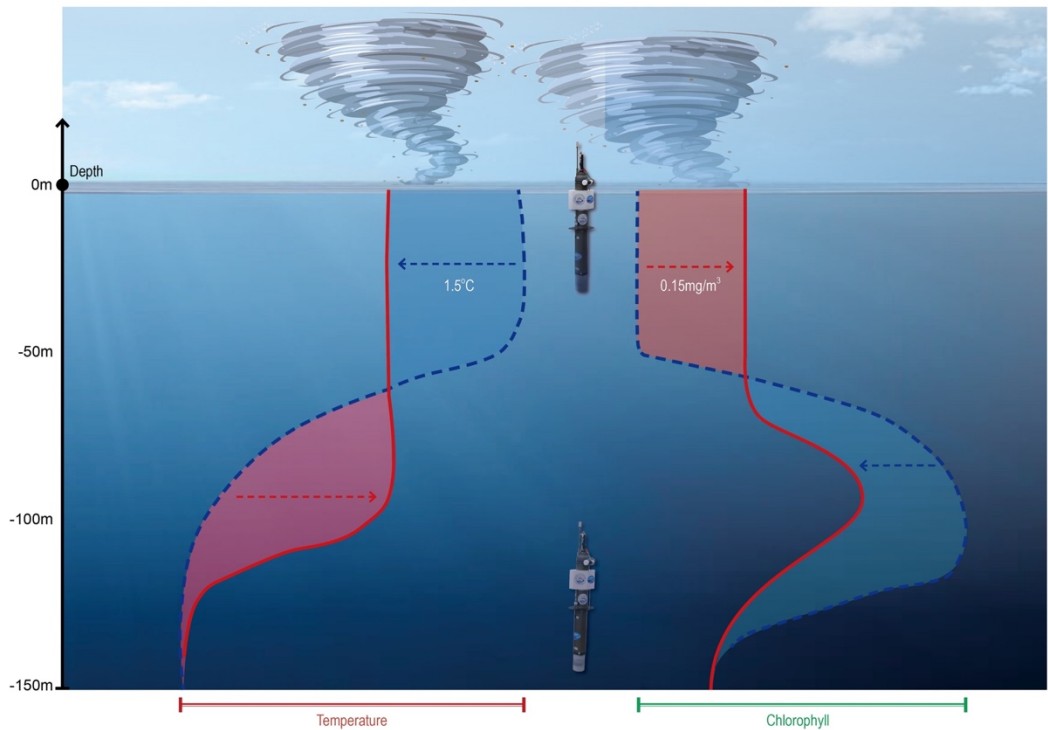

**Figure 5: Schematic diagram for a typhoon's impact on the vertical distribution of temperature (left) and chlorophyll (right). Blue dashed (red solid) lines correspond to vertical profiles before (after) a typhoon; blue/red dashed arrows mean decrement/increment of temperature and chlorophyll.**