# Peer review of "A Limited Effect of Sub-Tropical Typhoons on Phytoplankton Dynamics"

_Biogeosciences, 2020_

## Short Comment (SC1) · 30 Aug 2020

The general effect of tropical cyclones (TCs) on primary production/chlorophyll has also been studied at the global scale in Menkes et al. (2016). Looking at the effect of more than 1000 TCs, they reached the conclusion that the overall TC contribution to annual primary production was weak and amounted to 1%, except in a few limited areas (east Eurasian coast, South tropical Indian Ocean, Northern Australian coast, and Eastern Pacific Ocean in the TC-prone region) where it could locally reach up to 20–30% (Figure 1). These patterns were associated with the structure of the nutricline depth. While TCs could locally induce strong chlorophyll/primary production effects, the overall seasonally weak effect of TCs on primary production was explained by the limited regions of shallow nutricline. That contrasted with wider regions of shallow

thermoclines where TCs could induce an overall cooling on larger spatial scales on seasonal timescale.

Reference: Menkes, C.E., Lengaigne, M., Lévy, M., Ethé, C., Bopp, L., Aumont, O., Vincent, E., Vialard, J., Jullien, S., 2016. Global impact of tropical cyclones on primary production. Global Biogeochemical Cycles 30, 767–786. https://doi.org/10.1002/2015GB005214

[Figure]

[Figure]

**Fig. 1.** Spatial map of TC-induced primary production using 1000 TCs over 1998-2007 in percentage of the annual mean primary production. The contour of 0.5% is added in black (Menkes et al., 2016 Figure 10b)

---

## Referee Comment (RC1) · Peter Strutton (Referee) · 29 Sep 2020

This is a clear and concise study which debunks an idea that was the topic of a few papers a decade or two ago (I am thinking of a couple papers focussed on hurricanes off the US east coast or Gulf of Mexico). Those papers suggested, based on satellite chlorophyll, that hurricanes increase ocean productivity by stimulating mixing of nutrients. Of course satellite observations only show the surface story. The float observations presented here show the full subsurface variability. This paper shows that Typhoon Trami redistributed chlorophyll vertically, giving the appearance of increased surface chlorophyll, but in fact integrated chlorophyll and backscatter did not increase. The inclusion of backscatter is important here because it can address potential confounding issues of changes in chl:C.

The paper is fairly well written, but I think the native/proficient english speakers in the author team should pay more attention to the grammar. The first 4+ lines of the results section can be deleted because they repeat earlier text. There are other areas of repetition too (not quite verbatim). It seems like the discussion and conclusions could be shortened.

I have a few specific comments: Line 51: Upwelling isn't really mentioned in the rest of the paper, so either delete mention of it here or explain how it occurs, and follow up later. L88...: Delete "" around instrument model numbers. L102: Explain 'less than 300km' better. Is this a 300km x 300km box? A circle of radius 300km? L122: 10m intervals ... line 91 says 1m. L133: 0.18 and 0.15 are 0.13 and 0.08 higher than pre September 29. I don't understand.

Figures: It's difficult to see much of an increase in shallow chl in Fig 2b. The increase is more obvious in Fig 4b but that figure also made me wonder why there is an increase above the base of the mixed layer before Sep 30.
* * *

---

## Referee Comment (RC2) · Waldemar Walczowski (Referee) · 12 Oct 2020

General comments

The study raises the problem of the influence of tropical and sub-tropical cyclones on the primary production of the ocean. This type of work was mainly based on satellite data. The innovative aspect of the manuscript is the use of data from the biogeochemical Argo float (BGC Argo), in addition to satellite data. This method allowed the authors to draw new conclusions, most of them contrary to the previous works. Data from the upper 1000 m water column collected by the BGC Argo with a frequency of one day (measurements were made every night) allow observations of temporal evolution (and spatial changes) occurring near the sub-tropical typhoon Trami passage. Thanks to

these data, the authors can conclude that the observed enhancement of chlorophyll concentration in the surface layer is not the result of increased primary production (as previously thought only on the basis of satellite observations), but due to the displacement from deep layer of chlorophyll maxima to the surface. It is a well-written research paper with clearly defined assumptions and interesting, original, novel results. Construction of the manuscript is logical, paper is concise. The main weakness of the manuscript is the lack of reference to the Argo float position and trajectory. Autos say that 'Typhoon Trami passed over the BGC-Argo float'. Figure 1 clearly shows that the centre of the typhoon was approximately 60-100 nautical miles from the float position. Also the wind speed (Fig. 3a) shows that the float was not in the centre of the typhoon. This may not be relevant to the performed analysis, but should be explained in more detail than is done on line 209. The potential impact on the results of the spatial variability of the ocean properties should also be explained. The Argo float does not stay in place, it drifts. Significant changes in SST and chlorophyll content are also visible before the typhoon passes near the float (Fig. 2). Another weak point of the article are the repetitions, which the authors unfortunately did not avoid. Despite these weaknesses I consider that the manuscript is a valuable contribution to understanding the influence of cyclones (typhoons) on the ocean in general, and on primary production in particular. At the same time, the article shows the importance and usefulness of the Argo program. The use of BGC floats profiling with frequency higher than the commonly used 10 days gives additional possibilities to conduct research on short-term phenomena.

Specific comments

Lines 117-119 repeated information from lines 88-89 Lines 121-122 (float) 'was sampling daily from 1000 m depth to the surface at 10m intervals' Comment: what is the float measurements vertical resolution? In lines 90-91 you write 'Measurements were made every night (around 22:00 local time) to avoid in-vivo fluorescence non-photochemical quenching, with $\sim$ 1 m vertical resolution'.

Line 124: 'The BGC-Argo float profiles' Comment: There are 'sections' or 'section charts' at Fig 2a and 2b, not 'profiles' (see comment to Fig. 2 in technical corrections).

Line 133: 'at 0.18 and 0.15 mg chl a/m3, respectively. These increases represent changes of 0.13 and 0.08 mg chl a/m3, respectively, above the concentration measured on September 29 before the typhoon approached to the area.' Comment: Some mis-calculation. 0.18-0.15=0.03; 013-0.08=0.05. What was the concentration in September 29 ?

Line 148: 'The calculated profiles of temperature' Comment: same as in line 124.

Line 208, Figure 1. I am not sure if the method of representing the effects of a typhoon transition (Figure 1) is optimal. The averages for the 20-day period (September 10-September 30 and October 1- October 20) should strongly underestimate the effects of typhoon activity. Therefore, such significant temperature anomalies for September 28 and 29 are surprising. At the same time, no visible effects for dates before September 23.

Lines 208-210. What is the distance of the float to the typhoon centre ?

Lines 239-240: 'The results clearly show mixing is overwhelming the dynamics comparing with the upwelling' Comment: This is too general statement that should not be drawn from a single observation.

Technical corrections

I am not a native English speaker and I will not correct linguistic errors, yet in my opinion the article requires linguistic intervention. For example: lines 49-50 'Thus, strong typhoons, e.g., category 4 or 5, in mid-latitude regions are generally characterized as fast 50 moving and strong typhoons '. This sentence needs improvement (strong typhoons are strong typhoons). Figure 1. The float route should be showed (if the map scale allows). Figure 2. Title 'Profiles of temperature (a) and chlorophyll' is not correct. Figures 2a and 2b show not profiles but sections or section charts, while figures 2c and

2d show time series. The same remark applies to Figs 4a and 4b.

---

## Referee Comment (RC3) · Anonymous Referee #3 · 19 Oct 2020

**Comments on the manuscript "A Limited Effect of Sub-Tropical Typhoons on Phytoplankton Dynamics" by Chai, F., Wang, Y., Xing, X., Yan, Y., Xue, H., Wells, M., and Boss, E..**

General comment:

The manuscript "A Limited Effect of Sub-Tropical Typhoons on Phytoplankton Dynamics" by Fei Chai et al. describes the upper-ocean response, in terms of specific physical and biogeochemical features (temperature, mixed layer depth, chlorophyll, deep chlorophyll maximum), to the passage of Typhoon Trami (TT) offshore southern Japan coasts (Northwest Pacific Ocean). The issue has been already investigated in literature, recently showing that the overall role played by tropical cyclones on global primary production is quite limited (e.g. see Menkes et al., 2016, using ocean simulations). The novelty here is the use of high-frequency sampling vertical profiles of temperature and chlorophyll made available by a BGC-Argo float located near the Typhoon wake. BGC-Argo data significantly extend the amount of observations in comparison to what usually extracted from satellite, able to measure the surface in cloud-free conditions only. Conclusions show that mixing plays a larger role than upwelling, and TT weakly impacted on net primary production.

The manuscript is short and clear, with few but significant figures, very well explained and commented. However, I would point out some suggestions that may improve this study:

1. I think the paper would greatly increase its impact with some more deep investigation on the vertical mixing *vs* upwelling mechanism and the associated nutrient vertical flux, further supporting the thesis that no penetration through nutricline has effectively occurred. For example, would it be possible to include in the study some nutrient data (e.g. from a model, if not available from other sources) in order to fully demonstrate the typhoon impact as explained by the analysis of the BGC-Argo float measurements? As an example, data from EU Copernicus Marine Service could support the analysis of the physical driver (Global Analysis & Forecast Physics[1] at 1/12 degree), though not the same can be said for the biogeochemical parameters since the resolution at ¼ degree is possibly too coarse. I wonder whether Japan or China Ocean Forecasting operational centres may provide such model-derived data, or they can be available from other platforms. Another, probably more feasible, possibility would be to use a 1D-model approach, as the one developed by Terzic et al. (2019; https://doi.org/10.5194/bg-16-2527-2019) coupled with BFM biogeochemical model. In this direction, the study would surely benefit a lot from a model experiment which could reproduce the phenomenon and give the opportunity to deeply investigate the coupled physical-biogeochemical processes involved.

2. The paper focus is on typhoons, however, from the point of view of a reader, it would be interesting to know whether the results may be extended to all intense tropical cyclones (on the global scale) and which differences may be expected (also referring to literature) with extra-tropical cyclones.

3. Some typos and language editing is needed. Since I am not English mother-tongue I have only highlighted some points, but my feeling is that the paper readability would greatly benefit after a language editing.
* * *
[1]https://resources.marine.copernicus.eu/?option=com_csw&view=details&product_id=GLOBAL_ANALYSIS_FOREC AST_PHY_001_024

Specific comments:

1. L30: "an increase in the number of intense typhoons in the region" … how is quantified the intensity of typhoons? Readers of Biogeosciences may not be totally aware of typhoon intensity scale, so maybe a short comment can be added here. Further, the intensity classification has also been object of wide discussions (e.g. see Lei et al., 2017, https://www.sciencedirect.com/science/article/pii/S2225603218301589), so a clarification may be worth.

2. L38: "resulting in a negative response proposed to **facilitate** continued global temperature increase" … do you mean "support , sustain"? …not totally clear, please explain and rephrase.

3. L49: "e.g., category 4 or 5" … this may be clearer when specific comment n. 1 has been fulfilled.

4. L59-L68: the mechanism is clearly explained, though concisely. I think an illustrative sketch with mid-latitude / extra-tropical vs tropical regions would further help the reader to understand it.

5. L74: "It was suggested that the delayed response of surface chlorophyll is related to the growth time needed for phytoplankton to exploit the increased nutrient concentrations" … it would be interesting to explicitly add (or at least give a reference for) a time scale for the growth time.

6. L94: "Float data passed through a computer-based real-time quality control (RTQC)" some basic details about the RTQC would be helpful.

7. L100: "MODIS L3 daily data" please provide more info for this data here, not just in the Acknowledgements.

8. L106: "On September 30, a sublayer formed above the mixed layer." … where this information is extracted from? Fig.2 seems the right candidate, so you should refer to that one here.

9. L115-119: this information can be included in the "Methods" Section.

10. L133-136: a slight increase in surface Chla can be observed between 25 and 28 September, corresponding to weakening of the DCM chlorophyll intensity: can you comment on that? Moreover, the "reference baseline" of the surface Chla should be the one measured until 24 September, with almost constant values around 0.05 mg/m$^3$. Finally: any idea on the discrepancy between satellite and BGC-Argo following the second peak, later than 4 October?

11. L230: "The BGC-Argo floats typically provide three-dimensional observations at a 10-day profiling cycle to extend their operational lifetimes (Johnson and Claustre, 2016), a sampling frequency too low to capture synoptic weather and other short-term events." … Totally right, however BGC-Argo floats may also have shorter profiling cycles, e.g. 5-day (see Bittig et al., 2019; https://www.frontiersin.org/articles/10.3389/fmars.2019.00502/full).

Technical / other corrections:

1. L28: "the heat content in the upper ocean (with the sea surface temperature (SST) as the indicator)"… possibly: "the heat content in the upper ocean (**quantified by** sea surface temperature (SST) as an indicator)" or something similar.

2. L43: "The feedback from ocean to typhoon is important for the development and maintenance of typhoons, as **the** requires extracting energy from ocean surface" … maybe "it"?

3. L67: "thereby **transfer** new nutrients into the photic zone" … maybe "transfering"?

4. L70: "Besides the intensive wind field, typhoons are also **associating** with intensified rainfall and cloud" … maybe "associated"?

5. L71: "Satellite-based studies occasionally capture the ocean surface **feature** during the passage of typhoon and offer **more dataset** at the wake following typhoons" … maybe "features" and "more data" or "a richer dataset", or something similar?

6. L83: a short sentence closing the Introduction which states the object of the present work would be nice.

7. L95: "Data used in this study **are available at from** the Coriolis GDAC FTP server" … maybe "have been made available from" or simply "are available from"?

8. L108: MLT and MLC acronyms – though clear – have not been properly defined. You could simply say "We define MLT and MLC as …"

9. L127: "Figure 2a, **b**" … according to Fig.2, this should be Fig. 2a, c.

10. L132: "Figure 2b, **c**" … according to Fig.2, this should be Fig. 2b, d.

11. L193: "This is at least attribute to the solar radiation is much weaker comparing with tropics where the SST and stratification rebound quickly after passage of a typhoon" … this sentence should be corrected: "This is at least **attributed** to the solar radiation **which** is much weaker comparing with tropics where the SST and stratification rebound quickly after passage of a typhoon".

12. L208: two "indeed" in the same sentence, please correct.

13. L213: "The decreasing in SST is a general pattern", what do you refer to "general"? Do you maybe mean "well-known"?

14. L218: "The BGC-Argo measures vertical profiles that can be helpful to determine whether a net increasing in primary production, e.g., nutrient injection to upper ocean or subsurface bloom (Ye et al., 2013), taking place." … this sentence should be corrected: "The BGC-Argo  vertical profiles  can be helpful to determine whether a net increasing in primary production, e.g., nutrient injection to upper ocean or subsurface bloom (Ye et al., 2013), **takes** place."

15. L241: "redistribution of DCM over the mixed layer;" … I would say "redistribution of the DCM-localized chlorophyll content over the mixed layer" or something similar.

16. L242: "the delayed bloom that induced by typhoons may be due to the cloud coverage during the passage of typhoon. Thus, it implies an underestimation for the typhoon induced mixing and its associated vertical redistribution of water masses, while the impact of nutrients that being injected into euphotic zone can be overestimated." … this sentence should be corrected: "the delayed bloom  induced by typhoons may be due to the cloud coverage during the passage of typhoon. Thus, it implies an underestimation for the typhoon induced mixing and its associated vertical redistribution of water masses, while the impact of nutrients  being injected into euphotic zone can be overestimated."

17. L421: caption of Fig. 5 … blue dashed lines should correspond to vertical profiles before typhoon, red solid lines should correspond to vertical profiles at the typhoon passage on 30 September; blue/red dashed arrows mean decrement/increment (of T and Chla, values lacking in the 100m - layer) … please confirm and add to the caption.

References

Bittig, H.C., Maurer, T.L., Plant, J.N., Schmechtig, C., Wong, A.P.S., Claustre, H., Trull, T.W., Udaya Bhaskar, T.V.S., Boss, E., Dall'Olmo, G., Organelli, E., Poteau, A., Johnson, K.S., Hanstein, C., Leymarie, E., Le Reste, S., Riser, S.C., Rupan, A.R., Taillandier, V., Thierry, V. and Xing, X. (2019). A BGC-Argo Guide: Planning, Deployment, Data Handling and Usage. Front. Mar. Sci. 6:502. doi: 10.3389/fmars.2019.00502

Lei, X., Wong, W., and Fong, C. (2017). A challenge of the experiment on typhoon intensity change in coastal area. Tropical Cyclone Research and Review, 6(3-4), 94-97.

Menkes, C.E., Lengaigne, M., Lévy, M., Ethé, C., Bopp, L., Aumont, O., Vincent, E., Vialard, J., and Jullien, S. (2016). Global impact of tropical cyclones on primary production. Global Biogeochemical Cycles 30, 767–786. https://doi.org/10.1002/2015GB005214

Terzić, E., Lazzari, P., Organelli, E., Solidoro, C., Salon, S., D'Ortenzio, F., and Conan, P. (2019). Merging bio-optical data from Biogeochemical-Argo floats and models in marine biogeochemistry, Biogeosciences, 16, 2527–2542. https://doi.org/10.5194/bg-16-2527-2019

---

## Author Comment (AC1) · 12 Nov 2020

Interactive comment:

The general effect of tropical cyclones (TCs) on primary production/chlorophyll has also been studied at the global scale in Menkes et al. (2016). Looking at the effect of more than 1000 TCs, they reached the conclusion that the overall TC contribution to annual primary production was weak and amounted to 1%, except in a few limited areas (east Eurasian coast, South tropical Indian Ocean, Northern Australian coast, and Eastern Pacific Ocean in the TC-prone region) where it could locally reach up to 20–30% (Figure 1). These patterns were associated with the structure of the nutricline depth. While TCs could locally induce strong chlorophyll/primary production effects,

the overall seasonally weak effect of TCs on primary production was explained by the limited regions of shallow nutricline. That contrasted with wider regions of shallow thermoclines where TCs could induce an overall cooling on larger spatial scales on seasonal timescale.

Reference: Menkes, C.E., Lengaigne, M., LeÌvy, M., EtheÌ, C., Bopp, L., Aumont, O., Vincent, E., Vialard, J., Jullien, S., 2016. Global impact of tropical cyclones on primary production. Global Biogeochemical Cycles 30, 767–786. https://doi.org/10.1002/2015GB005214

Response: Thank you very much for providing the important reference. It is nice that our conclusion is consistent with the model result from Menkes et al. (2016), showing the limited contribution of typhoons on promoting net primary production in the Northwest Pacific Ocean. We have added the reference and discussion of underlying dynamics in the revised manuscript.

---

## Author Comment (AC2) · 13 Nov 2020

Referee Comment No.1:

This is a clear and concise study which debunks an idea that was the topic of a few papers a decade or two ago (I am thinking of a couple papers focused on hurricanes off the US east coast or Gulf of Mexico). Those papers suggested, based on satellite chlorophyll, that hurricanes increase ocean productivity by stimulating mixing of nutrients. Of course, satellite observations only show the surface story. The float observations presented here show the full subsurface variability. This paper shows that Typhoon Trami redistributed chlorophyll vertically, giving the appearance of increased surface chlorophyll, but in fact integrated chlorophyll and backscatter did not increase. The inclusion of backscatter is important here because it can address potential confounding issues of changes in Chl:C.

Dear Reviewer,

Thank you so much for evaluating our manuscript. We truly cherish the positive feedback and hope the study can be valuable to the community. Your suggestions were important to the improvement of our work. Please find detailed point-by-point responses below. We hope the revised manuscript and responses can ease your concerns. Please let us know if you have further comments.

The paper is fairly well written, but I think the native/proficient English speakers in the author team should pay more attention to the grammar. The first 4+ lines of the results section can be deleted because they repeat earlier text. There are other areas of repetition too (not quite verbatim). It seems like the discussion and conclusions could be shortened.

Response: Thank you very much. Following your suggestion, the repetition together with some others have been removed. Additionally, we have improved the discussion and conclusion more concisely.

I have a few specific comments:

Line 51: Upwelling isn't really mentioned in the rest of the paper, so either delete mention of it here or explain how it occurs, and follow up later.

Response: Thank you for the comment. Typhoons can simultaneously induce mixing and upwelling in the upper ocean, e.g., the mixing is related to intensive wind stress, while the upwelling is induced by the strong wind stress curl. The current study mainly describes the typhoon-induced mixing that results in the redistribution of chlorophyll in the upper ocean, and the impact of upwelling is much less pronounced. We totally agree with the reviewer that the description of mixing should be emphasized, while the description of upwelling can be toned down. In the revised manuscript, the upwelling related information is modified to emphasize the contrast between mixing and upwelling. We have tried to clarify the difference in these mechanisms in the revised text (L. 56-61).

L88...: Delete "" around instrument model numbers.

Response: Done. Thanks.

L102: Explain 'less than 300km' better. Is this a 300km x 300km box? A circle of radius 300km?

Response: Thank you for the suggestion. The description is improved as follows, 'Satellite observed information near BGC-Argo was calculated by spatially averaging over a surrounding circle with a radius of 300 km, excluding areas within 10 km of land.' (L. 116-118)

L122: 10m intervals ... line 91 says 1m.

Response: Thank you for pointing out the inconsistency. The vertical sampling frequency is indeed 1 m, which can be revealed from the detailed vertical structure (e.g., the MLD in Figure 2). We have

corrected the mistake (L. 105-106).

L133: 0.18 and 0.15 are 0.13 and 0.08 higher than pre-September 29. I don't understand.

Response: Thank you for noticing this issue. Different dates before the arrival of typhoon can be used as a reference. For example, the value a week before the typhoon, e.g., September 23, was 0.05 mg/m$^3$, and that the day right before the typhoon, e.g., September 29, was 0.07 mg/m$^3$. Thus, the change in chlorophyll on September 30 and October 3 compared with that on September 23 was 0.13 mg/m$^3$ and 0.1 mg/m$^3$, respectively. In addition, the change in chlorophyll compared with that on September 29 was 0.11 mg/m$^3$ and 0.08 mg/m$^3$, respectively. We apologize for mixing up the differences, and we have updated the numbers using September 29 as the reference to make them consistent throughout the text (L. 163-166).

Figures: It's difficult to see much of an increase in shallow Chl in Fig 2b. The increase is more obvious in Fig 4b but that figure also made me wonder why there is an increase above the base of the mixed layer before Sep 30.

Response: Thank you for asking. We agree with the reviewer that the change in shallow Chl can hardly be distinguished from Figure 2b, because the figure is mainly describing the prominent decrease of subsurface chlorophyll maximum (SCM). The change in surface Chl is clearly captured in Figure 2d, which is further compared with the satellite observations. On the other hand, the increase in Chl above the base of the mixed layer (ML) is actually related to the deepening of the ML, which is induced by increasing wind (Figure 3a). The climatological depth of the ML for the study site, derived from the World Ocean Atlas (WOA; Locarnini et al., 2018) following the method of Kara et al. (2000), is 30 m, 46 m and 66 m in August, September and October, respectively. Thus, the elevated subsurface Chl is at least related to the gradual deepening of the ML (Figure 2b), which simultaneously increases the subsurface temperature (Figure 4a). The process is different from the typhoon induced dynamics that tends to decrease temperature and increase Chl at the surface while increase temperature and decrease Chl at the subsurface. Moreover, the movement of BGC-Argo, though it travels a small distance, can end up in a different environmental, which can be captured in the profile. In particular, BGC-Argo was moving northward during the study period and the background information varied slightly. The corresponding information is added to the discussion (L. 149-154).

Reference:

Kara, A. B., P. A. Rochford & H. E. Hurlburt. 2000. An optimal definition for ocean mixed layer depth. J. Geophys. Res., 105(C7), 16803-16821.

Locarnini, R.A., A. V. Mishonov, O. K. Baranova, T. P. Boyer, M. M. Zweng, H. E. Garcia, J. R. Reagan, D. Seidov, K. Weathers, C. R. Paver & I. Smolyar. 2018. World Ocean Atlas 2018, Temperature. A. Mishonov Technical Ed.; NOAA Atlas NESDIS 81, 1, 52 pp.

---

## Author Comment (AC4) · 13 Nov 2020

Referee Comment No.2:

General comments

The study raises the problem of the influence of tropical and sub-tropical cyclones on the primary production of the ocean. This type of work was mainly based on satellite data. The innovative aspect of the manuscript is the use of data from the biogeochemical Argo float (BGC Argo), in addition to satellite data. This method allowed the authors to draw new conclusions, most of them contrary to the previous works. Data from the upper 1000 m water column collected by the BGC Argo with a frequency of one day (measurements were made every night) allow observations of temporal evolution (and spatial changes) occurring near the sub-tropical typhoon Trami passage. Thanks to these data, the authors can conclude that the observed enhancement of chlorophyll concentration in the surface layer is not the result of increased primary production (as previously thought only on the basis of satellite observations), but due to the displacement from deep layer of chlorophyll maxima to the surface. It is a well-written research paper with clearly defined assumptions and interesting, original, novel results. Construction of the manuscript is logical, paper is concise.

Dear Reviewer,

Thank you so much for evaluating our manuscript. We are glad the study can add new perspectives to typhoon induced ecosystem dynamics. The pointed weaknesses by the reviewer are important in improving the manuscript. We have modified the draft accordingly and hope the revised manuscript can fully meet your expectations. Please let us know if you have further comments.

The main weakness of the manuscript is the lack of reference to the Argo float position and trajectory. Autos say that 'Typhoon Trami passed over the BGC-Argo float'. Figure 1 clearly shows that the centre of the typhoon was approximately 60-100 nautical miles from the float position. Also, the wind speed (Fig. 3a) shows that the float was not in the centre of the typhoon. This may not be relevant to the performed analysis, but should be explained in more detail than is done on line 209. The potential impact on the results of the spatial variability of the ocean properties should also be explained. The Argo float does not stay in place, it drifts. Significant changes in SST and chlorophyll content are also visible before the typhoon passes near the float (Fig. 2).

Response: Thank you for the suggestion. Indeed, the typhoon center passed the study area approximately 100 km to the left of the BGC-Argo. Because of the intensive wind pattern during its passage, the typhoon-induced responses are dominant and can impact the surrounding area up to a few hundred kilometers away (Wang, 2020). As the reviewer mentioned, the distance between the typhoon and BGC-Argo is less influential for conducting the analysis.

Additionally, because of the movement of BGC-Argo, it undergoes different environmental conditions that are illustrated in Figure R1. Specifically, the BGC-Argo was located at 133.1°E, 30°N on Sep. 25, 133°E, 30.7°N on Sep. 30, and 133.3°E, 31.1°N on Oct. 3; thus, the zonal movement is not prominent, while the meridional shift is approximately 120 km northward. The labeled position in Figure 1 represents the location of BGC-Argo when typhoon passing over on September 30. The climatological mixed layer depth (MLD) for both the north and the south location along its trajectory can be obtained from the World Ocean Atlas (WOA; Locarnini et al., 2018) following Kara et al. (2000). The MLD to the north is 30 m, 46 m and 66 m on August, September and October, respectively, while that to the south is 28 m, 41 m and 60 m, correspondingly. Thus, the MLD along the trajectory of BGC-Argo is increasing, though the change is not prominent, during the study period, and it can result in the elevated subsurface Chl. This is why we think the change in subsurface Chl is related to the seasonal evolution of MLD and enhanced wind stress. Corresponding information has been added in the discussion in the revised manuscript (L. 149-154).

[Figure]

Figure R1. The averaged mixed layer depth (meter) in September and October overlaid with the trajectories of typhoon (cyan color, from Sep. 20 to Oct. 1) and BGC-Argo float (blue color, from Sep. 23 to Oct. 3). The location of float on Sep. 30, corresponding to the location shown in Figure 1, is labeled as a green dot.

Another weak point of the article are the repetitions, which the authors unfortunately did not avoid.

Response: Thank you very much. Following your suggestion, the manuscript has been checked throughout and repetitions have been removed. Additionally, we have improved the discussion and conclusion for a more precise presentation.

Despite these weaknesses I consider that the manuscript is a valuable contribution to understanding the influence of cyclones (typhoons) on the ocean in general, and on primary production in particular. At the same time, the article shows the importance and usefulness of the Argo program. The use of BGC floats profiling with frequency higher than the commonly used 10 days gives additional possibilities to conduct research on short-term phenomena.

Thank you very much for the positive evaluations. Though there are some weaknesses that can hardly been overcome because of the limited observations, our study offers a unique opportunity to improve the understanding of typhoon-induced dynamics. In addition, the sampling frequency is critical for a more comprehensive characterization of intensive and rapid processes.

Specific comments

Lines 117-119 repeated information from lines 88-89 Lines 121-122 (float) 'was sampling daily from 1000 m depth to the surface at 10m intervals' Comment: what are the float measurements vertical resolution? In lines 90-91 you write 'Measurements were made every night (around 22:00 local time) to avoid in-vivo fluorescence non- photochemical quenching, with ~ 1 m vertical resolution'.

Response: Thank you for pointing out the inconsistency. The vertical sampling frequency is indeed 1 m, which can be observed from the detailed vertical structure (e.g., the MLD in Figure 2). We have corrected the mistake (L. 105-106).

Line 124: 'The BGC-Argo float profiles' Comment: There are 'sections' or 'section charts' at Fig 2a and 2b, not 'profiles' (see comment to Fig. 2 in technical corrections).

Response: Thanks. The word has been modified accordingly.

Line 133: 'at 0.18 and 0.15 mg Chl a/m$^3$, respectively. These increases represent changes of 0.13 and 0.08 mg Chl a/m$^3$, respectively, above the concentration measured on September 29 before the typhoon approached to the area.' Comment: Some mis- calculation. 0.18-0.15=0.03; 013-0.08=0.05. What was the concentration in September 29?

Response: Thank you for noticing this issue. Different dates before the arrival of a typhoon can be used as a reference. For example, the value on September 23, a week prior to the typhoon, was 0.05 mg/m$^3$, while the value one day before the typhoon on September 29 was 0.07 mg/m$^3$. Thus, the change in chlorophyll on September 30 and October 3 compared with that on September 23 was 0.13 mg/m$^3$ and 0.1 mg/m$^3$, respectively. In addition, the change in chlorophyll compared with that on September 29 was 0.11 mg/m$^3$ and 0.08 mg/m$^3$, respectively. We apologize for mixing up the differences, and we have updated the numbers using September 29 as the reference to make them consistent throughout the text (L. 163-166).

Line 148: 'The calculated profiles of temperature' Comment: same as in line 124.

Response: Done. Thanks.

Line 208, Figure 1. I am not sure if the method of representing the effects of a typhoon transition (Figure 1) is optimal. The averages for the 20-day period (September 10- September 30 and October 1- October 20) should strongly underestimate the effects of typhoon activity. Therefore, such significant temperature anomalies for September 28 and 29 are surprising. At the same time, no visible effects for dates before September 23.

Response: This is a great question. Typhoon-induced ocean surface responses vary depending on typhoon features and the ocean status beneath. Composite methods are usually applied to obtain a general pattern for typhoon-induced ocean changes (e.g., Lin et al., 2017; Wang, 2020). For typhoon Trami, the wind speed is weak during the first half of its lifespan, e.g., before Sep. 25; thus, the typhoon-driven changes in the ocean are small. Between Sep. 25 and Sep. 29, the typhoon is moving very slowly and inducing prominent sea surface cooling and chlorophyll enhancement, which is captured in Figure 1. After Sep. 29, the typhoon is still strong but fast-moving, and the situation is favorable to induce upper ocean responses, though the change is smaller compared with that before Sep. 29. It will be ideal to have observations along the track where the largest changes are being identified, but BGC-Argo data are still very sparse, and the only dataset available is enclosed in current study. Fortunately, the typhoon-induced changes are very clear and can be useful to help understand the dynamics.

Lines 208-210. What is the distance of the float to the typhoon centre?

Response: Thank you for asking. The typhoon is passing over BGC-Argo from the left side at a smallest distance of approximately 100 km. The information is added (L. 148-149).

Lines 239-240: 'The results clearly show mixing is overwhelming the dynamics comparing with the upwelling' Comment: This is too general statement that should not be drawn from a single observation.

Response: Thank you for the comment. We agree with the reviewer that the observation from a single BGC-Argo cannot guarantee a general conclusion. The statement has been modified to tone down the comparison between mixing and upwelling, only focusing on current study (L. 287-288).

Technical corrections

I am not a native English speaker and I will not correct linguistic errors, yet in my opinion the article requires linguistic intervention. For example: lines 49-50 'Thus, strong typhoons, e.g., category 4 or 5, in mid-latitude regions are generally characterized as fast 50 moving and strong typhoons '. This sentence needs improvement (strong typhoons are strong typhoons). Figure 1. The float route should be showed (if the map scale allows). Figure 2. Title 'Profiles of temperature (a) and chlorophyll' is not correct. Figures 2a and 2b show not profiles but sections or section charts, while figures 2c and 2d show time series. The same remark applies to Figs 4a and 4b.

Response: Thank you for this suggestion. The language has been modified by a native English speaker for the entire manuscript. In addition, the inaccurate words are corrected. The movement of the float is mostly limited in a very small area (Figure R2); thus, its trajectory is not shown in the figures. Alternatively, the description for BGC-Argo locations is added in detail in the revised manuscript (L. 149-152).

[Figure]

Figure R2. The averaged mixed layer depth (meter) in September and October overlaid with the trajectories of typhoon (cyan color, from Sep. 20 to Oct. 1) and BGC-Argo float (blue color, from Sep. 23 to Oct. 3).

Reference:

Kara, A. B., P. A. Rochford & H. E. Hurlburt. 2000. An optimal definition for ocean mixed layer depth. Journal Geophysical Research, 105(C7), 16803-16821.

Lin, S., W. Z. Zhang, S. P. Shang & H. S. Hong. 2017. Ocean response to typhoons in the western North Pacific: Composite results from Argo data. Deep-Sea Res. I, 123, 62-74.

Locarnini, R. A., A. V. Mishonov, O. K. Baranova, T. P. Boyer, M. M. Zweng, H. E. Garcia et al. 2018. World Ocean Atlas 2018, Vol. 1: Temperature. A. Mishonov Technical Ed.; NOAA Atlas NESDIS 81, 52 pp.

Wang, Y. 2020. Composite of typhoon induced sea surface temperature and chlorophyll-a responses in the South China Sea, Journal Geophysical Research: Oceans, 125, e2020JC016243.

---

## Author Comment (AC5) · 13 Nov 2020

Referee Comment No.3:

General comment:

The manuscript "A Limited Effect of Sub-Tropical Typhoons on Phytoplankton Dynamics" by Fei Chai et al. describes the upper-ocean response, in terms of specific physical and biogeochemical features (temperature, mixed layer depth, chlorophyll, deep chlorophyll maximum), to the passage of Typhoon Trami (TT) offshore southern Japan coasts (Northwest Pacific Ocean). The issue has been already investigated in literature, recently showing that the overall role played by tropical cyclones on global primary production is quite limited (e.g. see Menkes et al., 2016, using ocean simulations). The novelty here is the use of high-frequency sampling vertical profiles of temperature and chlorophyll made available by a BGC-Argo float located near the Typhoon wake. BGC-Argo data significantly extend the amount of observations in comparison to what usually extracted from satellite, able to measure the surface in cloud-free conditions only. Conclusions show that mixing plays a larger role than upwelling, and TT weakly impacted on net primary production.

The manuscript is short and clear, with few but significant figures, very well explained and commented. However, I would point out some suggestions that may improve this study.

Dear Reviewer,

Thank you so much for providing the positive feedback on our manuscript; your endorsement is highly encouraging. We truly appreciate the valuable comments, which are important for improving the manuscript. We hope the revised manuscript fully meet your expectations. Please let us know if you have further comments.

1. I think the paper would greatly increase its impact with some more deep investigation on the vertical mixing *vs* upwelling mechanism and the associated nutrient vertical flux, further supporting the thesis that no penetration through nutricline has effectively occurred. For example, would it be possible to include in the study some nutrient data (e.g. from a model, if not available from other sources) in order to fully demonstrate the typhoon impact as explained by the analysis of the BGC-Argo float measurements? As an example, data from EU Copernicus Marine Service could support the analysis of the physical driver (Global Analysis & Forecast Physics at 1/12 degree), though not the same can be said for the biogeochemical parameters since the resolution at 1⁄4 degree is possibly too coarse. I wonder whether Japan or China Ocean Forecasting operational centres may provide such model-derived data, or they can be available from other platforms. Another, probably more feasible, possibility would be to use a 1D-model approach, as the one developed by Terzic et al. (2019; https://doi.org/10.5194/bg-16-2527-2019) coupled with BFM biogeochemical model. In this direction, the study would surely benefit a lot from a model experiment which could reproduce the phenomenon and give the opportunity to deeply investigate the coupled physical-biogeochemical processes involved.

Response: Thanks for the comment. The current study focuses on the use of a high-frequency vertical observation to study the impact of a typhoon on the upper ocean, e.g., temperature and chlorophyll. It will be ideal to have the nutrient observations from BGC-Argo, but unfortunately, the BGC-Argo was not equipped with nitrogen sensor. On the other hand, no net change was identified for chlorophyll and bbp within the top 150 m (Figure 3d), indicating that there was no net production being generated in the upper ocean. Following your suggestion, we have investigated the typhoon-induced nutrient changes by combining reanalysis data from HYCOM, climatological profiles from the World Ocean Atlas (WOA) and obtained BGC-Argo data in the current study. However, the HYCOM reanalysis is not capturing the typhoon-induced changes during the passage of typhoon (Figure R1). The climatological nitrogen profile in September and October shows low value (< 2 μmol/kg) within the upper ocean till 125m, corresponding to the nutricline depth. Thus, in this case, the typhoon-induced mixing cannot introduce nutrients into the upper ocean, and the redistribution of nutrients within the mixed layer cannot stimulate the net growth of phytoplankton. Our conclusion is consistent with the model result from Menkes et al. (2016), showing limited contribution of typhoons to promoting net primary production. The suggested reference is very helpful to guide future studies. As such, we will employ a numerical model to simulate

comprehensively upper ocean processes during typhoon passage as typhoon-induced changes may vary considerably across individual cases, depending on typhoon characteristics and oceanic conditions. Nevertheless, it is important to note that the main conclusion of this study (i.e., limited contribution of typhoons to promoting net primary production) is important but may not be applicable to all cases of typhoons.

[Figure]

Figure R1. The sea surface temperature (SST) averaged for the period (left) before, e.g., between Sep. 16 and Sep. 30, and (middle) after, e.g., between Oct. 1 and Oct. 15, the passage of typhoon Trammi whose trajectory is shown in cyan color. (right) The change of SST, e.g., the averaged SST after typhoon minus the averaged SST before typhoon.

2. The paper focus is on typhoons, however, from the point of view of a reader, it would be interesting to know whether the results may be extended to all intense tropical cyclones (on the global scale) and which differences may be expected (also referring to literature) with extra-tropical cyclones.

Response: Thank you for the suggestion. Indeed, the current study focuses on a single typhoon based on the unique BGC-Argo observations. The result is representative of a general situation and can be applicable for other tropical cyclones. We have added a general discussion and conclusion on the application of the acquired information. However, it is important to point out the typhoon-induced ocean responses can vary depending on the feature of the typhoon, e.g., the strength and translation speed, and the status of the upper ocean including the mixed layer depth, stratification, nutrients and chlorophyll. There are some classic (Babin et al., 2004) and recent studies (Lin and Oey, 2016; Lin et al., 2017; Wang, 2020) that investigated the prototypical oceanic responses to tropical cyclones by compositing different tropical cyclones. Their results can be representative for delineating the typhoon-induced changes, since the investigated scenario is highly representative in typhoon intensity and trespassing environment. Corresponding text has been added in the revised manuscript (L. 256-264, 283-290).

3. Some typos and language editing are needed. Since I am not English mother-tongue I have only highlighted some points, but my feeling is that the paper readability would greatly benefit after a language editing.

Response: Thank you very much. Following your suggestion, we have requested language editing by a native English speaker. We hope the improved manuscript can ease your concern.

Specific comments:

1. L30: "an increase in the number of intense typhoons in the region" ... how is quantified the intensity of typhoons? Readers of Biogeosciences may not be totally aware of typhoon intensity scale, so maybe a short comment can be added here. Further, the intensity classification has also been object of wide discussions (e.g. Lei et al., 2017, www.sciencedirect.com/science/article/pii/S2225603218301589), so a clarification may be worth.

Response: Thank you for the suggestion. The information for typhoon intensity is added in the revised manuscript, along with the associated reference (L. 125-131).

2. L38: "resulting in a negative response proposed to **facilitate** continued global temperature increase" ... do you mean "support, sustain"? ...not totally clear, please explain and rephrase.

Response: Thank you for the comment. Because of increased stratification resulted from global warming, the typhoon can hardly elevate ocean primary production that can relax the warming by absorbing carbon dioxide (He and Soden, 2015). The reduced primary production accelerates the warming trend. The 'negative response' can lead to misunderstanding for readers, and we have removed it and rephrased the sentence to improve the clarification (L. 36-41).

3. L49: "e.g., category 4 or 5" ... this may be clearer when specific comment n. 1 has been fulfilled.

Response: Thank you. The description for the intensity of the typhoon is incorporated in the revised manuscript following your suggestion (please see response to Specific Comment No. 1 above).

4. L59-L68: the mechanism is clearly explained, though concisely. I think an illustrative sketch with mid-latitude / extra-tropical vs tropical regions would further help the reader to understand it.

Response: Thank you for the suggestion. The majority of BGC-Argos currently in the water are in the Antarctic Ocean. In the future, we plan to deploy more BGC-Argos in the mid-latitude and tropical western Pacific, and we hope they can offer a more comprehensive description for the typhoon's impact in regions with different stratification, vertical structure of nutrients among others.

5. L74: "It was suggested that the delayed response of surface chlorophyll is related to the growth time needed for phytoplankton to exploit the increased nutrient concentrations" ... it would be interesting to explicitly add (or at least give a reference for) a time scale for the growth time.

Response: Thank you for the comment. The typical growth rates of phytoplankton, e.g., the time required to double the biomass, is around few days, depending on the species of phytoplankton. Previous studies revealed a diatom bloom 3 days after the passage of a typhoon in the Northwest Pacific Ocean (Pan et al., 2017). In addition, substantial works based on satellite observations indicated that the chlorophyll peak happens between 5 and 7 days after a typhoon, which was suggested to be the time required for phytoplankton accumulation (Wang, 2020). Corresponding information has been added in the revised manuscript (L. 81-83).

6. L94: "Float data passed through a computer-based real-time quality control (RTQC)" some basic details about the RTQC would be helpful.

Response: Thank you. The float data was quality controlled following the requirement of the BGC-Argo Program (Schmechtig et al., 2016), and this information has been added to the manuscript (L. 108-112).

7. L100: "MODIS L3 daily data" please provide more info for this data here, not just in the Acknowledgements.

Response: Done. The information is added in the method section (L. 114-118).

8. L106: "On September 30, a sublayer formed above the mixed layer." ... where this information is extracted from? Fig.2 seems the right candidate, so you should refer to that one here.

Response: Thank you very much for the comment. Actually, the sublayer is not influential on the calculation of MLT and MLC, thus the related text is removed in the revision.

9. L115-119: this information can be included in the "Methods" Section.

Response: Deeply appreciate your nice catch. There is indeed some duplication between the pointed results and method. We have removed the description for Argo sampling from the result. Thank you.

10. L133-136: a slight increase in surface Chla can be observed between 25 and 28 September, corresponding to weakening of the DCM chlorophyll intensity: can you comment on that? Moreover, the "reference baseline" of the surface Chla should be the one measured until 24 September, with almost constant values around 0.05 mg/m$^3$. Finally: any idea on the discrepancy between satellite and BGC-Argo following the second peak, later than 4 October?

Response: Thank you for asking; this is definitely a good question. The slight increase in Chl is related to the seasonal evolution of the mixed layer (ML), which is deepening associating with increasing wind (Figure 3a). In the current study, BGC-Argo is moving northward by 120 km from Sep. 20 to Oct. 3 (Figure R2). Specifically, BGC-Argo locates at 133.1°E, 30°N on Sep. 25, 133°E, 30.7°N on Sep. 30, and 133.3°E, 31.1°N on Oct. 3. As BGC-Argo approaches the coast, the vertical distribution of chlorophyll changes with weakening of the DCM and an increasing MLD. The climatological MLD, which is derived from the World Ocean Atlas (WOA; Locarnini et al., 2018) following the method of Kara et al. (2000), for the float location in the north is 30 m, 46 m and 66 m on August, September and October, respectively, while that in the south is 28 m, 41 m and 60 m, correspondingly. Thus, the movement of BGC-Argo, though it travels a small distance, can end up in a different environmental, which can be captured in the sections. The elevated surface Chl is at least related to the gradual deepening of the ML (Figure 2b), which simultaneously increases the surface temperature (Figure 4a). The process is different from the typhoon induced dynamics that tends to decrease temperature and increase Chl at the surface while increase temperature and decrease Chl at the subsurface. The corresponding information is added to the discussion (L. 149-154, 188-190, 248-252).

[Figure]

Figure R2. The averaged mixed layer depth (meter) in September and October overlaid with the trajectories of typhoon (cyan color, from Sep. 20 to Oct. 1) and BGC-Argo float (blue color, from Sep. 23 to Oct. 3). The right panel is showing an enlarged region showing the trajectory of the float, with the location on Sep. 30 labeled in green, corresponding to the location shown in Figure 1.

The reference is important for determining the responses of the upper ocean. In this study, we used the original time series as much as possible, but we needed to use the anomaly for acquiring typhoon-induced changes, e.g., Figure 1 and 4. Because the longevity of typhoon spans from Sep. 21 to Oct. 1, a 20 day average before (Sep. 10 to Sep. 30) and after (Oct. 1 to Oct 20) the typhoon is used to capture the pre- and post-typhoon situation for the entire region. For the location of BGC-Argo, only a 7-day average, e.g., Sep. 13 to 19, is applied as the reference. This is because the typhoon can induce ahead-of-eye cooling (Glenn et al., 2016) and the ocean surface responses can take place much earlier before its arrival (Figure 2c; Wang, 2020). Because the obtained anomaly in temperature and chlorophyll are close to zeros (Figure 4c), the different is less prominent for the period from Sep. 13 to 19 and from Sep. 19 to 24. Thus, the reference is defined using the period much earlier.

The observed difference in chlorophyll after October 2 is believed to be related to the limited observations of the satellite. The typhoon can induce strong rainfall associated with large area being covered by cloud (Lin and Oey, 2016); thus, the available satellite data are often sparse. In addition, the coastal region is characterized by high chlorophyll and can still impact the region, though the area less than 10 km from the coast is excluded. As more satellite observations become available, the difference between satellite and BGC-Argo data may be less pronounced on October 10 or later (Figure 2d).

11. L230: "The BGC-Argo floats typically provide three-dimensional observations at a 10-day profiling cycle to extend their operational lifetimes (Johnson and Claustre, 2016), a sampling frequency too low to capture synoptic weather and other short-term events." ... Totally right, however BGC-Argo floats may also have shorter profiling cycles, e.g. 5-day (see Bittig et al., 2019; www.frontiersin.org/articles/10.3389/fmars.2019.00502/full).

Response: Excellent point. Your suggestion is absolutely consistent with the message we hope to deliver in the current study, along with typhoon-induced mixing. The sampling frequency is critical, as different profiling cycles can result in various conclusions. For example, the applied BGC-Argo is sampling one cycle per day, which can resolve the peak of chlorophyll on Sep. 30. However, if we adjust the sampling frequency to once every 5 days, the likelihood of missing the peak would be 80%. We have discussed this in-depth in one of our recent publications on GRL (Xing et al., 2020), and a consistent statement is added in the revised manuscript. Even so, we now refer to float cycling times of 5-10 days (L. 275-277).

Technical / other corrections:

1. L28: "the heat content in the upper ocean (with the sea surface temperature (SST) as the indicator)" ... possibly: "the heat content in the upper ocean (**quantified by** sea surface temperature (SST) as an indicator)" or something similar.

Response: Done.

2. L43: "The feedback from ocean to typhoon is important for the development and maintenance of typhoons, as **the** requires extracting energy from ocean surface" ... maybe "it"?

Response: Done.

3. L67: "thereby **transfer** new nutrients into the photic zone" ... maybe "transferring"?

Response: Done.

4. L70: "Besides the intensive wind field, typhoons are also **associating** with intensified rainfall and cloud" ... maybe "associated"?

Response: Done.

5. L71: "Satellite-based studies occasionally capture the ocean surface **feature** during the passage of typhoon and offer **more dataset** at the wake following typhoons" ... maybe "features" and "more data" or "a richer dataset", or something similar?

Response: Done.

6. L83: a short sentence closing the Introduction which states the object of the present work would be nice.

Response: A brief summary for the introduction and the object of the paper is added (L. 93-95).

7. L95: "Data used in this study **are available at from** the Coriolis GDAC FTP server" ... maybe "have been made available from" or simply "are available from"?

Response: Done.

8. L108: MLT and MLC acronyms – though clear – have not been properly defined. You could simply say "We define MLT and MLC as ..."

Response: Thanks. The definitions are added (L. 135-143).

9. L127: "Figure 2a, **b**" ... according to Fig.2, this should be Fig. 2a, c.

10. L132: "Figure 2b, **c**" ... according to Fig.2, this should be Fig. 2b, d.

Response: Thanks for the suggestion. We have modified the text when referring to sub-panels in Figure 2 (L. 158, 163).

11. L193: "This is at least attribute to the solar radiation is much weaker comparing with tropics where the SST and stratification rebound quickly after passage of a typhoon" ... this sentence should be corrected: "This is at least **attributed** to the solar radiation **which** is much weaker comparing with tropics where the SST and stratification rebound quickly after passage of a typhoon".

Response: Done.

12. L208: two "indeed" in the same sentence, please correct.

Response: Done.

13. L213: "The decreasing in SST is a general pattern", what do you refer to "general"? Do you maybe mean "well-known"?

Response: Done.

14. L218: "The BGC-Argo measures vertical profiles that can be helpful to determine whether a net increasing in primary production, e.g., nutrient injection to upper ocean or subsurface bloom (Ye et al., 2013), taking place." ... this sentence should be corrected: "The BGC-Argo  vertical profiles  can be helpful to determine whether a net increasing in primary production, e.g., nutrient injection to upper ocean or subsurface bloom (Ye et al., 2013), **takes** place."

Response: Done.

15. L241: "redistribution of DCM over the mixed layer;" ... I would say "redistribution of the DCM-localized chlorophyll content over the mixed layer" or something similar.

Response: Done.

16. L242: "the delayed bloom that induced by typhoons may be due to the cloud coverage during the passage of typhoon. Thus, it implies an underestimation for the typhoon induced mixing and its associated vertical redistribution of water masses, while the impact of nutrients that being injected into euphotic zone can be overestimated." ... this sentence should be corrected: "the delayed bloom  induced by typhoons may be due to the cloud coverage during the passage of typhoon. Thus, it implies an underestimation for the typhoon induced mixing and its associated vertical redistribution of water masses, while the impact of nutrients  being injected into euphotic zone can be overestimated."

Response: Done.

17. L421: caption of Fig. 5 ... blue dashed lines should correspond to vertical profiles before typhoon, red solid lines should correspond to vertical profiles at the typhoon passage on 30 September; blue/red dashed arrows mean decrement/increment (of T and Chla, values lacking in the 100m - layer) ... please confirm and add to the caption.

Response: Thank you very much. Corresponding information has been added in the caption.

References

Bittig, H.C., Maurer, T.L., Plant, J.N., Schmechtig, C., Wong, A.P.S., Claustre, H., Trull, T.W., Udaya Bhaskar, T.V.S., Boss, E., Dall'Olmo, G., Organelli, E., Poteau, A., Johnson, K.S., Hanstein, C., Leymarie, E., Le Reste, S., Riser, S.C., Rupan, A.R., Taillandier, V., Thierry, V. and Xing, X. (2019). A BGC-Argo Guide: Planning, Deployment, Data Handling and Usage. Front. Mar. Sci. 6:502. doi: 10.3389/fmars.2019.00502

Lei, X., Wong, W., and Fong, C. (2017). A challenge of the experiment on typhoon intensity change in coastal area. Tropical Cyclone Research and Review, 6(3-4), 94-97.

Menkes, C.E., Lengaigne, M., Lévy, M., Ethé, C., Bopp, L., Aumont, O., Vincent, E., Vialard, J., and Jullien, S. (2016). Global impact of tropical cyclones on primary production. Global Biogeochemical Cycles 30, 767–786. https://doi.org/10.1002/2015GB005214

Terzić, E., Lazzari, P., Organelli, E., Solidoro, C., Salon, S., D'Ortenzio, F., and Conan, P. (2019). Merging bio-optical data from Biogeochemical-Argo floats and models in marine biogeochemistry, Biogeosciences, 16, 2527–2542. https://doi.org/10.5194/bg-16-2527-2019

Reference:

Babin, S. M., J. A. Carton, T. D. Dickey & J. D. Wiggert. 2004. Satellite evidence of hurricane induced phytoplankton blooms in an oceanic desert. Journal Geophysical Research, 109, C03043.

Glenn, S., T. Miles, G. Seroka, Y. Xu, R. Forney, F. Yu et al. 2016. Stratified coastal ocean interactions with tropical cyclones. Nature Communication, 7(1), 10887-10887.

He, J., & B. J. Soden. 2015. Anthropogenic weakening of the tropical circulation: The relative roles of direct $CO_2$ forcing and sea surface temperature change. Journal of Climate, 28(22), 8728–8742.

Kara, A. B., P. A. Rochford & H. E. Hurlburt. 2000. An optimal definition for ocean mixed layer depth. Journal Geophysical Research, 105(C7), 16803-16821.

Lin, S., W. Z. Zhang, S. P. Shang & H. S. Hong. 2017. Ocean response to typhoons in the western North Pacific: Composite results from Argo data. Deep-Sea Research Part I, 123, 62-74.

Lin, Y. C. & L. Y. Oey. 2016. Rainfall-enhanced blooming in typhoon wakes. Scientific Reports, 6, 31310.

Locarnini, R. A., A. V. Mishonov, O. K. Baranova, T. P. Boyer, M. M. Zweng, H. E. Garcia et al. 2018. World Ocean Atlas 2018, Vol. 1: Temperature. A. Mishonov Technical Ed.; NOAA Atlas NESDIS 81, 52 pp.

Pan, G., F. Chai, D. Tang & D. Wang. 2017. Marine phytoplankton biomass responses to typhoon events in the South China Sea based on physical-biogeochemical model. Ecological Modelling, 356, 38-47.

Schmechtig, C., V. Thierry & The Bio Argo Team, 2016. Argo quality control manual for biogeochemical data. https://doi.org/10.13155/40879

Wang, Y. 2020. Composite of typhoon induced sea surface temperature and chlorophyll-a responses in the South China Sea, Journal Geophysical Research: Oceans, 125, e2020JC016243.

Xing, X., M. L. Wells, S. Chen, S. Lin & F. Chai. 2020. Enhanced Winter Carbon Export Observed by BGC-Argo in the Northwest Pacific Ocean. Geophysical Research Letter, e2020GL089847.